# Data compression of Bridge Resilience Control: Algorithm and case analysis

**Ming Chen** (ID) *

School of Architecture Engineering, Shanghai ZhongQiao Vocational and Technical University, Shanghai, China

* chenmchen1975@126.com

## Abstract

Bridge inspection and structural health monitoring represent the primary approaches to managing bridge resilience. Data acquired through inspection and monitoring activities provides an effective technical basis for the systematic implementation of bridge resilience control strategies. Yet, uninterrupted monitoring and diverse inspection campaigns have yielded an enormous volume of data, which directly imposes comprehensive and stringent challenges on data storage, transmission and processing. Consequently, data compression has become a research priority in the field of bridge resilience control. However, existing data compression algorithms are all general-purpose data processing techniques, which decouple the intrinsic physical relevance between monitoring data and bridge structural behaviors. To tackle this limitation, this study integrates domain knowledge, the time-series characteristics of bridge monitoring data, and bridge deterioration models into the design of a novel data compression algorithm. This approach addresses the issue of indiscriminate data compression inherent to conventional algorithms, thereby enabling efficient data compression while preserving critical bridge structural state information. By incorporating domain knowledge, the proposed method transforms raw monitoring data into data information with engineering attributes. based on these attributes, a set of interrelated monitoring data is further converted into a small subset of key data that is directly applicable to bridge resilience control practice. Leveraging the steady-state variation law of bridge operational performance, the dynamic structural characteristics of bridges are extracted from time-series monitoring data, which correspondingly reduces the storage demand of time-series datasets. For data sampling intervals interrupted by various types of system faults, a sparse data supplementation method is proposed. After data supplementation, the complete dataset is further refined by utilizing the inherent time-series characteristics of the monitoring data, which not only ensures data integrity but also further reduces the overall data volume. Simulation analyses demonstrate that the domain knowledge-based compression method achieves a data compression ratio of 75%. Moreover, the comprehensive compression ratio exceeds 92% after the synergistic processing of time-series feature

**Data availability statement:** The runnable python code for bridge structural simulation and data compression is deposited in Zenodo (DOI: 10.5281/zenodo.18080198) under the MIT Open Source License. The code repository includes a detailed README document that describes all required dependencies (e.g., MATLAB R2023b, Python 3.9), installation steps and execution instructions to facilitate reproducibility.

**Funding:** The author(s) received no specific funding for this work.

**Competing interests:** The authors have declared that no competing interests exist.

extraction and sparse data supplementation, with a data fidelity rate of 95%. These performance metrics indicate that the proposed method can reduce the data storage costs and transmission bandwidth consumption associated with bridge resilience control by 75% to 92%. Meanwhile, the 95% feature retention accuracy satisfies the engineering precision requirements for bridge resilience control assessments, which effectively reconciles the inherent contradiction between data compression efficiency and structural evaluation accuracy.

## 1. Introduction

### 1.1 Background

Bridges constitute the core components of urban infrastructure systems, and their structural resilience plays a decisive role in the overall resilience of urban engineering systems. Conventional structural detection, structural health monitoring (SHM), and other technical methodologies have exerted a fundamental role in the resilience assessment and operational maintenance management of bridge structures. Nevertheless, with the rapid advancement of science and engineering technologies, bridge inspection and real-time structural monitoring are generating an enormous volume of multi-source heterogeneous data streams. Although such datasets contain comprehensive information reflecting the actual service status of bridges, they also impose severe challenges and heavy burdens on the efficiency of data storage and real-time transmission in engineering applications. Taking the bridge health monitoring system as a typical instance, various types of sensors are deployed for continuous data acquisition via high-frequency sampling protocols. Uncompressed raw monitoring data rapidly depletes storage resources, which inevitably results in a substantial escalation in capital expenditure pertaining to hardware capacity expansion and long-term maintenance.

By using data compression technology to reduce the volume of data, the deployment and maintenance costs of servers and cloud storage will be significantly reduced. At the same time, it supports the full lifecycle data archiving of bridges, providing a data foundation for long-term performance evolution analysis of bridges. In the field of bridge resilience control, commonly used algorithms can be divided into three categories: lossless compression, lossy compression, and time-series data-specific compression. Lossless compression algorithms include Huffman coding, the LZW algorithm, and the DEFLATE algorithm. Huffman encoding is based on data frequency allocation of encoding length. The LZW algorithm achieves adaptive encoding by dynamically constructing a dictionary. The DEFLATE algorithm combines LZ77's sliding window matching with Huffman encoding. The typical characteristic of non-computational compression is that the reconstructed data is completely identical to the original data.

The bridge lossy compression algorithm improves compression rate by discarding non-critical information. Typical lossy compression algorithms include Discrete Cosine Transform (DCT), Wavelet Transform, and Principal Component Analysis

(PCA). DCT converts time-series data into frequency-domain data while retaining low-frequency trend components; the Wavelet transform takes into account both time-series and frequency-domain local features, and can capture data mutation signals; PCA achieves high-dimensional data dimensionality reduction through orthogonal transformation and is often used as a preprocessing step in intelligent algorithms. The specialized compression algorithm for time-series data is designed for monitoring data with strong correlation and periodicity, including differential encoding, rotary gate algorithm (SDT), and online piecewise linear approximation (OLA). Differential encoding encodes the residuals of adjacent data, often combined with entropy encoding. The revolving door algorithm is based on a threshold, fits data with a line, and saves the inflection point. OLA algorithm divides data segments in real time, which is suitable for online compression tasks of edge computing nodes.

With the rapid development of artificial intelligence technology, based on the core requirements of structural disaster resistance, recovery and adaptation, intelligent algorithms integrate traditional compression principles with artificial intelligence, edge computing and other technologies to achieve data dimensionality reduction and key feature retention on the premise of ensuring the accuracy of resilience assessment, providing core support for real-time early warning, disaster response and long-term operation and maintenance. The first research direction of intelligent algorithms is the adaptive optimization results of classical algorithms, such as the adaptive revolving gate algorithm based on LSTM network optimization threshold, and the wavelet transform optimization method that integrates a genetic algorithm to screen wavelet basis functions. The second research direction is based on AI-driven deep compression technology. The research content includes an auto-encoder that integrates high compression rate and damage feature preservation with key resilience evaluation indicators as constraints, compressed sensing that reduces the sampling and transmission pressure of sensing nodes based on data sparsity, a lightweight neural network compression scheme adapted to edge deployment based on quantization, pruning, and other technologies, combined with domain adaptive principal component analysis, and graph neural network fusion compression based on bridge structure topology. These intelligent algorithms can adapt to engineering scenarios such as extreme disaster monitoring, long-term resilience assessment, real-time edge warning, and multi-source data fusion. The goal of the algorithm is to transmit disaster mutation data with low power consumption and store historical data with a high compression ratio, in order to support performance degradation analysis, reduce node resource consumption, and accurately evaluate bridge resilience.

## 1.2 Motivation and innovation

The existing research results on data compression provide technical support for data compression in the field of bridge resilience control, especially with the rapid development of intelligent algorithms, greatly improving the level of data compression. However, there are still some key technologies that need to be broken through in the field of data compression for bridge resilience control.

Firstly, the general data compression algorithm does not involve knowledge in the field of bridges, and does not take into account the mechanical properties of bridges and the physical meaning of data, which may result in the loss of key structural state information in compressed data.

Second, bridges themselves and the monitoring data collected during their operational time exhibit considerable inherent redundancy, yet existing general-purpose compression algorithms fail to effectively leverage this redundancy for data compression purposes.

Thirdly, due to the limitations of the collection equipment and extreme weather, the data collected for bridge resilience control also has the characteristic of sparsity, which requires data filling to be effectively applied.

To address the aforementioned issues, research has been conducted on the following aspects:

Firstly, a data compression method based on knowledge in the field of bridge resilience control has been proposed, which reduces the geometric data volume of bridges.

Secondly, a time-series data compression method has been proposed. This method focuses on data within the coverage range of the time step window, achieving dynamic compression of time-series data.

Thirdly, a method based on sparse data filling of pre- and post-dataset is proposed, which supplements sparse data in the time-series and refines the supplementary data, solving the problem of data loss in the field of bridge rebound control.

## 2. Related work

Bridge health monitoring is the main technical means of bridge resilience control. The research and application of bridge health monitoring began in the 1970s. After the 1990s, with the rapid development of large-scale bridge construction, bridge health monitoring systems have been widely applied. However, the application of a large number of data collection methods has led to an unusually large amount of data obtained for bridge health monitoring, prompting data compression processing to become the main research direction for bridge health monitoring.

Traditional data compression methods have been widely used in the data processing of bridge health monitoring systems, including wavelet transform, Fourier transform, PCA, Empirical Mode Decomposition, Huffman Encoding, SAX, and their fusion applications. Wavelet transform is a mathematical framework for multi-scale decomposition using adjustable scale wavelet functions. Its core principle is to decompose signals into components of different scales and compress them by removing redundant information. For example, Lorenzo Bernardini et al. [1] proposed a bridge damage detection method based on driving vibration. By using the wavelet transform to extract the time-frequency characteristics of vibration signals, this method can maintain a damage recognition accuracy of over 95% even when the data compression rate reaches 85%. The Fourier transform is a mathematical tool that converts time-domain signals into frequency-domain signals. Its core is to decompose any periodic or non-periodic signal that satisfies the Dirichlet condition into a superposition of sine/cosine waves of different frequencies, amplitudes, and phases. The Fourier transform is suitable for data compression of periodic, stationary signals, and the compressed data can be directly applied to feature extraction. For example, Premjeet Singh et al. [2] proposed a method that integrates natural excitation techniques and empirical Fourier decomposition to analyze environmental bridge vibration data and determine the modal parameters of the bridge. This method can provide an accurate and robust estimation of bridge modal parameters. PCA is an unsupervised data dimensionality reduction and feature extraction method that maps high-dimensional data to a low-dimensional space through linear transformation, reducing redundancy while preserving the main information of the data. Principal component analysis can eliminate the data correlation between sensors in bridge health monitoring systems and is suitable for large-scale multi-source data compression, with a data compression rate of over 90% [3]. EMD is an adaptive signal decomposition method that decomposes complex non-stationary and nonlinear signals into several stationary and physically meaningful intrinsic mode functions and a residual component. The empirical mode decomposition method does not require prior knowledge and has strong adaptive ability. This method is suitable for bridge health monitoring in complex environments, with a data compression rate of up to 50% while reducing signal noise [4]. Huffman encoding is a lossless data compression algorithm that reconstructs data that is completely identical to the original data after compression. This method can solve the problem of low spatial efficiency caused by unpredictable outliers in the compression process of time series data [4]. SAX is a data dimensionality reduction and symbolic representation method for time series. Its core is to convert continuous time series into discrete symbol streams, which can significantly compress data volume while preserving key features. After receiving real-time perception data in the monitoring system, SAX is applied to compress the data, and then efficient classification tasks are performed based on the compressed data to complete the evaluation of bridge structure status [4]. In order to integrate the advantages of traditional compression algorithms, scholars have fused multiple algorithms and applied them to data processing in the field of bridge health monitoring. For example, Zhou L et al. [5] proposed a scheme that combines complementary set empirical mode decomposition, wavelet threshold denoising, and PCA fusion to achieve data reduction while denoising. Zhang Feng Yuan et al. [6] proposed a compression method combining wavelet transform and LZW encoding, and applied multiple sets of actual sampling data for simulation testing. The results showed that the compression ratio was better than 10.8.

The advantages of traditional data compression algorithms are mature principles, strong hardware adaptability, and high accuracy in data restoration after compression. However, traditional data compression algorithms are limited to simple data compression, and their compression process and feature extraction process are separated, requiring a redesign of the feature extraction algorithm for bridge structures. With the rapid development of artificial intelligence technology, the application of machine learning in the field of bridge resilience control has received widespread attention from scholars and the engineering community. Common machine learning methods include Bayesian, CNN, LSTM, Transformer, GNN, Reinforcement learning, etc. The Bayesian method is a statistical inference framework based on probability theory, whose core is to update the knowledge of unknown parameters through prior probabilities and observation data, and output a posterior probability distribution. Kullaa Jyrki [7] combined Bayesian theory with virtual sensing technology to solve the problem of dense sensor networks and repeated collection of vibration data, generating a large amount of data that needs to be stored. CNN is suitable for processing data with a grid structure. In bridge resilience control, the combination of CNN's convolutional layer, pooling layer, and fully connected layer is used to extract data features and compress the original data [8]. LSTM excels at capturing long-term dependencies of time-series data. LSTM is often combined with convolutional networks, generative adversarial networks, etc., for bridge monitoring data compression, which can reduce data volume and ensure the accuracy of subsequent structural state evaluation. LSTM is a special RNN architecture that is designed with gating mechanisms and cell states at its core. LSTM can map high-dimensional raw data to a low-dimensional space while preserving its implicit structural features, thereby reducing data transmission volume [9]. Transformer is a deep learning model proposed by the Google team, which uses a self-attention mechanism to directly calculate the correlation weight between each position in the sequence and all other positions, achieving the function of capturing global dependencies and complex correlations of data [10]. Transformer uses Embedding technology to transform high-dimensional data into fixed-dimensional feature vectors, which can significantly compress data and save storage space and data transmission. GNN is suitable for processing graph-structured data, adept at capturing topological relationships between data, achieving feature extraction, and data compression. GNN is commonly used to identify defects such as cracks, concrete spalling, and steel corrosion in bridge structures, and can compress high-dimensional data into low-dimensional data [11]. The advantages of reinforcement learning are dynamic adaptive decision-making and multi-objective optimization. Through the interaction learning between the agent and the environment, redundant information is eliminated, achieving dimensional compression of high-dimensional data [12]. The research on machine learning algorithms focuses on the accuracy of data compression, but neglects the physical requirements of bridge resilience control, and has shortcomings such as data dependence and poor interpretability. Improvement is needed to establish the correlation between machine learning results and the mechanical principles of bridge structures.

During the operational time of bridges, both bridge inspection and bridge health monitoring obtain data with time-series characteristics. Therefore, the compression of time-series data is an important aspect of bridge resilience control. In existing literature, there are relatively few studies that directly focus on compressing time-series data of bridges, and the vast majority of the literature focuses on feature extraction of bridges. The process of extracting bridge features is also the process of compressing time-series data. Time-series data compression algorithms are divided into two categories: lossless compression and lossy compression. Lossless compression is a reversible compression technique that achieves 100% restoration of compressed data by identifying and eliminating statistical redundancy in the data [13]. The core principle of lossy compression is to actively discard redundant or secondary information in the data that has no significant impact on the target application scenario in exchange for a higher compression ratio, allowing irreversible information loss in the compression and decompression process. Lossy compression is the main method for processing bridge time-series data. At present, research on lossy compression algorithms for bridge temporal data is focused on the field of machine learning. On the basis of maintaining the status information of the bridge, scholars have integrated multiple machine learning algorithms to compress the temporal data of the bridge. For example, combining CNN with BiGRU can extract data features while identifying temporal features between data, achieving data compression [14]. In order to predict the

degradation of high-strength steel wire in long-term service, Long Xiao et al. [15] proposed a hybrid prediction model that integrates the sparrow search algorithm and LSTM. Simulation analysis shows that the algorithm has significant advantages in both convergence speed and optimization accuracy. Wang Ziyi et al. [16] integrated the grey wolf optimization algorithm and LSTM, using the grey wolf optimization algorithm to synergistically optimize the hyper-parameters of LSTM, while integrating temporal feature extraction and signal decomposition techniques. The case analysis results indicate that, compared to CNN and LSTM, the algorithm combining the grey wolf optimization algorithm and LSTM is more suitable for bridge displacement prediction tasks. In addition, time-series data compression algorithms in other fields also have certain reference value for the field of bridge engineering. For example, in order to solve the problem of the relatively lagging development of computing power and storage capacity in high-performance computing platforms in computational fluid dynamics, Adalberto Perez et al. [17] introduced Gaussian process regression under the Bayesian framework, which can achieve posterior recovery of initially discarded information. Research has confirmed that this method is not only suitable for compressing three-dimensional turbulent spatial field data, but also for compressing discrete time series datasets. In the field of the Internet of Things, efficient data compression technology is crucial for reducing storage costs and improving query performance. Due to its high precision and wide dynamic range, floating-point sequential data poses great challenges to data compression. To address this issue, Wenjing Wang et al. [18] proposed a numerical pattern-aware compression algorithm for floating-point time-series data. This algorithm introduces a classification model at the time window level to identify hidden numerical patterns in the data, and constructs a two-layer decision architecture to achieve a balance between compression ratio and time overhead. Johannes Pöppelbaum et al. [19] proposed a novel quaternion temporal data compression method based on neural network models. This method first divides long-term data into several data segments, extracts the minimum, maximum, mean, and standard deviation of each data segment as representative features, and encapsulates these features into quaternions to generate quaternion numerical time series data. In the field of cloud-based digital twin systems, monitoring key performance indicators is the core link to ensure system security and reliability. However, the monitoring data generated by the system is massive, and data compression technology has become a necessary means to save data transmission bandwidth and storage space. Zicong Miao et al. [20] proposed a collaborative compression method for multivariate temporal data based on a two-step compression scheme. This method first performs a morphology-based clustering algorithm to group multivariate temporal data; Subsequently, it optimizes the compressive sensing technology to achieve collaborative compression of grouped data. The experimental results show that this method can achieve efficient data compression while effectively preserving the complex temporal correlations between indicators: at a compression ratio of 30%, the root mean square error of the correlation between reconstructed data and original data is only 0.0489.

## 3. Related definitions

### 3.1 Data

Data is the fine-grained information of bridge structures, and the data for bridge toughness control is defined as equation (1)

$$d_i = < l_i, t_i, v_i, r_i, k_i, w_i > \tag{1}$$

In equation (1), $d_i$ is the data. $l_i$ is the label of the data, which has uniqueness in bridge resilience control. $t_i$ is the type of data. $v_i$ is the value of the data. $r_i$ is the associated information of the data. $k_i$ is the knowledge information that constrains the data. $w_i$ is the data weight used to represent the importance of data $d_i$ in data compression, $w_i \in [0, 1)$. For data with different knowledge constraints, there are significant differences in the value of $w_i$. Taking the section in bridge structures as an example, for rectangular sections, the width data $b$ and height data $h$ have the same importance when calculating the inertia moment of the section. However, for I-shaped sections, the importance of flange thickness and

 

web height is significantly higher than other data. The dataset consisting of all data for bridge resilience control is defined as equation (2).

$$D = \{d_1, d_2, \ldots\ldots, d_n\}$$

(2)

The dataset represented by equation (2) is a collection of various types of data for bridge resilience control. The amount of data contained in this set increases with increased operational time throughout the entire life cycle of the bridge. The dataset is the direct object of Data compression research.

## 3.2 Knowledge

Knowledge is the core element that distinguishes different application fields. The knowledge in the field of bridges is defined as equation (3)

$$k_i =< kt_{i,k}, kr_{i,s}, kv_{i,v} >$$

(3)

In equation (3), $kt_{i,k}$ is the identifier for knowledge $k_i$. $kr_{i,s}$ is the relationship between the value $v_i$ of data $d_i$ and the knowledge value $kv_{i,v}$, $kr_{i,s} \in \{>, <, ==, \geq, \leq, \ldots\ldots\}$. $kv_{i,v}$ is the value of knowledge $k_i$, which comes from various codes or experiences. Therefore, the determination of data $d_i$ compliance does not need to be obtained through training of large-scale models, which can significantly reduce the training workload of large-scale models. In this study, knowledge was applied to large-scale models through knowledge templates. Different data correspond to different domain knowledge, and the corresponding knowledge templates are also different. The cross-sectional knowledge template is defined as equation (4)

$$T = \left\{ \left( d_i \bowtie d_{i+1} \bowtie \ldots \right), \left( d_j \bowtie d_{j+1} \bowtie \ldots \right), \ldots \right\}$$

(4)

In equation (4), $\bowtie$ represents the knowledge association between data. For example, in the cross-sectional knowledge template, $\bowtie$ is the angle between the data and the $x-\text{axis}$.

## 3.3 Correlation

Correlation refers to the correlation between data. The correlations studied in this article include geometric correlations and temporal correlations. Geometric correlation refers to the geometric shapes or structures composed of data within the subset $D_k$ of a dataset, $D_k \subset D$. When $D_k == D$, $D_k$ constitutes the entire bridge structure. Time-series correlation refers to the correlation generated in bridge structure over time. For example, the acceleration data and displacement data collected by the bridge health monitoring system all satisfy time-series correlation.

 **3.3.1 Geometric correlation.** Based on geometric correlation, data can be used to establish sections, components, substructures, etc. Geometric correlation is defined as equation (5)

$$r_i = (d_i, rt_i, rp_i, d_j, d_{j+1} \ldots\ldots)$$

(5)

In equation (5), $d_i$ represents the data involved in the correlation with $r_i$. $rt_i$ is the correlation type, for cross-sectional data $rt_i == \widetilde{S}$. $rp_i$ is the correlation parameter. In the geometric correlation of cross-sections, $rp_i = \{m_{i,1} : p_{i,1}, m_{i,2} : p_{i,2}, \ldots\ldots\}$. $m_{i,j}$ is the key of the correlation, used to represent the attributes of the correlation. $p_{i,j}$ is the value of the correlation, used to characterize the specific function of the correlation. $d_j$ is the data associated with the influence of $r_i$. $d_i$ is related to the engineering characteristics and correlation attributes, and can be one (section) or multiple (substructure or structure). By applying geometric correlations, data can be integrated and evolved into sections, components, substructures, and structures.

**3.3.2 Time-series correlation.** Time-series correlation represents the variation of data over time. Time-series correlation can be continuous or discrete, with equal or unequal intervals. Time-series correlation is defined as equation (6).

$$r_i = (rt_i, rt_t)$$

(6)

In equation (6), $rt_i$ represents the correlation type, which can be acceleration, displacement, strain, etc. $rt_t$ is the time of data acquisition. Time-series correlation can characterize the full lifecycle changes of certain data during the operational time of bridge structures, and is the most direct information for recording the continuous changes in bridge operation status.

## 4. Data compression methods

The goal of Data compression is to reduce the amount of data required for low bridge resilience control supported by large models. It mainly includes three aspects: first, knowledge-based compression which mainly deals with the compression of data with geometric correlations in bridges. The second is based on time-series compression, which mainly deals with the compression of data with time-series correlations. The third is sparse data compression, which mainly deals with the compression of incomplete data in bridge resilience control.

### 4.1 Data compression based on domain knowledge

The significant difference between bridge resilience control data and conventional data lies in its domain engineering properties. The application of these engineering properties helps to significantly reduce the amount of data required for large-scale model training and improve model training efficiency.

**4.1.1 Weight setting.** The main problem solved by intelligent algorithms in existing large-scale models is to find the optimal solution, and the selection of weights is one of the core components of intelligent algorithms. Knowledge based Data compression applies domain knowledge to weight adjustment, avoiding the traditional method of using a large amount of data to train and adjust weights, which can significantly reduce the amount of training data. For example, in the process of refining cross-sectional data, each data $d_i$ has a corresponding importance coefficient $w_i$. Based on $w_i$, the weights of the large-scale model algorithm can be adjusted to accelerate convergence speed. Taking the commonly used neural network in large-scale models as an example, its output function and squared error are given by equations (7) and (8), respectively.

$$y_i = f(\xi_i) = \sum_{j=0}^{L} \omega_{ji} x_{ji}$$

(7)

In equation (7), $y_i$ is the output of the neural network. $f$ is the activation function, $\omega_{ji}$ is the connection weight between the $j$th input neuron and the $i$-th output neuron, and $L$ is the number of all input neurons.

$$e(\omega) = \frac{1}{2} \sum_{i=1}^{J} (h_i - y_i)^2$$

(8)

In equation (8), $h_i$ is the expected output value, and $J$ is the number of output neurons. Using gradient descent optimization algorithm, the weight update rule is equation (9)

$$\omega_{ji} = \omega_{ji} - \eta_{ji} x_{ji} (y_i - h_i)$$

(9)

In equation (9), $\eta_{ji}$ is the learning factor, which is generally a constant (positive value). Therefore, the selection of $\eta_{ji}$ becomes the key to the efficiency and quality of large-scale model training. Smaller $\eta_{ji}$ values have lower training efficiency, while larger $\eta_{ji}$ values may not achieve the optimal solution.

In the process of data compression based on domain knowledge, $\eta$ is designed as a variable whose value is limited by the importance coefficient $w_j$ corresponding to $d_j$. Taking the I-shaped cross-section as an example, the learning factor is shown in equation (10).

$$\eta_{ji} = -0.2 \times \frac{d}{d\xi}\left(\frac{1 - e^{\xi}}{1 + e^{\xi}}\right)$$

(10)

$$\xi = 1 - (d_j(w_j))$$

(11)

In equation (11), $d_j(w_j)$ is used to extract the importance coefficient of the data $d_j$. The physical meaning of equation (10) is that a smaller learning factor is adopted for the data with higher importance, while a larger learning factor is adopted for the data with lower importance, so as to improve the training efficiency of the large model.

**4.1.2 Algorithm.** The difference between knowledge-based Data compression algorithms and traditional algorithms lies in the application domain knowledge setting learning factor $\eta_{ji}$. At the same time, domain knowledge is applied to convert data $d_i$ into bridge data with engineering properties.

Taking the I-shaped section as an example, $d_i$ stores data such as flange width, thickness, and web height, which can be combined into an I-shaped section through domain knowledge. A neural network model is invoked in the proposed algorithm. This neural network model is a fully connected deep neural network. The input layer consists of 4 feature dimensions; the hidden layer is composed of three layers with the ReLU activation function adopted, and the output layer contains a single neuron with a linear activation function applied. In the process of model training, adaptive training is realized by means of early stopping and learning rate decay, which not only ensures the convergence performance of the model, but also mitigates overfitting. Besides, the differentially weighted features are applied throughout the entire training process. The mean squared error (MSE) is selected as the loss function for this neural network. This loss function has excellent compatibility with the Adam optimizer, which enables stable gradient calculation and fast convergence speed. Meanwhile, the square term in the mean squared error imposes a heavier penalty on large errors, thus driving the model to prioritize the correction of samples with large deviations.

**Algorithm: Bridge Section Data Closed-Loop Processing and Dataset Update Algorithm**
```
// Input: D – Original bridge structure dataset; T – Cross-sectional data template
// Output: D – Updated bridge structure dataset
Algorithm BridgeSectionDataProcessing(D, T)
    D_c = ∅:
    for each d_i in D do
        if d_i is cross-sectional data then
          current_d = d_i
          while current_d.r ≠ ∅ do
            Add current_d to D_c
            current_d = current_d.r
          end while
          Add current_d to D_c
        end if
    end for
    D̄ = NeuralNetworkModel(D_c)
    for each d̄_i in D̄ do
        Create new data point d̄_k
```

```
      d̄_k.v = d̄_i
      Add d̄_k to D
    end for
    RecursiveRemove(D, d_i, d_i.l)
    return D
end Algorithm
```

## 4.2 Compression of time-series data

The data for bridge resilience control includes real-time data collected by the bridge health monitoring system. The typical characteristic of this type of data is continuous cyclic sampling, with an exceptionally large volume of data that requires a significant amount of computing and communication resources. The method of time-series data compression is to analyze and transform the data within the window period, and then replace the entire window period with the center point data.

### 4.2.1 Definition and transformation of time-series data.

(1) Displacement mode dataset

The definition of displacement mode dataset is shown in equation (12).

$$A = \{a_1, a_2, \ldots\ldots, a_n\} \tag{12}$$

In equation (12), $a_i$ is the displacement mode value analyzed based on the $i$-th sensor. $a_i$ is an ordered set that is time-dependent. $n$ is the number of sensors in the collection system. The acquisition time of $a_1, a_2, \ldots\ldots, a_n$ is synchronized.

$$a_i = \{g_{k,i}, g_{k,i+1}, \cdots\cdots, g_{k,i+z}\} \tag{13}$$

In equation (13), $g_{k,i}$ is the displacement mode value at time $t_0$, and $g_{k,i+z}$ is the displacement mode value at time $t_z$. The format $g_{k,i}, g_{k,i+1}, \cdots$ is equation (1).

(2) Window Size

Time-series data compression refers to the compression processing of data within the window period. The size of the window is defined as the window size, denoted by $W$. When $W$ is small, the data loss is relatively small while the compression ratio is low; when $W$ is large, the compression ratio is high, whereas the data loss is relatively large. The window size is correlated with the operational time of the bridge and abrupt incident during its operation. According to the nonlinear model for natural deterioration of concrete proposed in Ref. [21], it is expressed as equation (14).

$$S(\theta) = \begin{cases} 95 & 0 \leq \theta \leq 1 \\ 95 - 0.066(\theta - 1)^2 & \theta \geq 2 \end{cases} \tag{14}$$

In equation (14), $S(\theta)$ denotes the bridge technical condition in the $\theta$-th year, and the value of 95 represents the initial condition score of the bridge. The initial value of the window size can be selected according to the operational time of the bridge based on equation (14). The initial value of $W$ is calculated by equation (15) as follows:

$$W = \lceil S(\theta) \rceil \tag{15}$$

(3) Data transformation

Data transformation is the transformation of time-series data for bridge resilience control based on engineering properties. After data transformation, the engineering significance of the data becomes clearer and the process of Data compression can be simplified. For example, using equation (16) to transform the displacement mode of multiple acquisition points at the same time into a curvature mode

$$\overline{\varphi}_i = \frac{\varphi_{i-1} - 2\varphi_i + \varphi_{i+1}}{l^2}$$

(16)

In equation (16), $\overline{\varphi}_i$ is the curvature mode at measurement point $i$. $\varphi_i$ is the displacement mode at measurement point $i$. $l$ is the distance between two measuring points. By data transformation, the displacement mode obtained by the acquisition system is converted into the curvature mode of each measuring point.

**4.2.2 Time-series Data compression algorithm.** Applying Data compression algorithms to process curvature modes within the window and replacing time-series data with center point data can reduce the amount of system data. The steps of Data compression are as follows.

**Algorithm: Curvature Mode Data Compression Based on GIN Classification**
```
// Input:
// - G: Curvature mode dataset (raw input)
// - θ: Bridge operational time (parameter)
// Output: D̄ - Compressed curvature mode
Algorithm TimeSeriesDataCompression(G, θ)
      D̄ = ∅
      W = ⌈S(θ)⌉
      for each D_{t_i}^W in D do
          g_{t_i} = GenerateCurvatureModeDiagram(D_{t_i}^W)
          Add g_{t_i} to G
          X = CallGINClassification(G)
          while X > 1 do
            W = W / 2 // Halve the window step size
            X = CallGINClassification(G)
          end while
          sum_dj = 0
          for j from 1 to W do
            sum_dj = sum_dj + d_j
          end for
d̄ = sum_dj / W
          Add d̄ to D̄
          Remove d_j from G
      end for
      if G ≠ ∅ then
          goto Step 3 // Re-enter loop if unprocessed data remains in G
      end if
      return D̄
end Algorithm
```

This algorithm transforms displacement modes into curvature modes and applies a large-scale image classification algorithm to classify curvature mode maps within a time step. When the number of classifications $X = 1$, it is considered that the curvature mode data at each time point can be refined, and the mean of all curvature modes is taken as the representative value of the refined data.

In the above algorithms, the GIN model is invoked to classify the graphs generated by the curvature mode. When the number of categories $X = 1$, the data in this category is compressed by taking the mean value of the data within the

category. When the number of categories $X > 1$, half of the window size is used to perform reclassification. Meanwhile, the data fidelity rate is set to 95% in the program design.

### 4.3 Sparse data compression

Affected by factors such as unexpected equipment failures and extreme weather, the data acquired by the information acquisition system for bridge resilience control may suffer from missing sampling, leading to incomplete data information [22–24]. Directly discarding sparse data will reduce the continuity of bridge resilience control, and fail to effectively characterize the continuous variation of the bridge's operational performance throughout its whole life cycle. To effectively utilize these sparse data, the sparse data is first supplemented in this study, and then compressed with the time series data compression algorithm introduced in Section 4.2.2. The methods for data supplementation are as follows:

The compensation window size for missing data is set as $W_{md}$ in accordance with Equation (15). Let the missing data set be $D^{t_k} = \left\{ d_1^{t_k}, d_2^{t_k}, \cdots\cdots, d_n^{t_k} \right\}$, where $d_i^{t_k}$ denotes the missing value of the $i$-th sensor, $n$ represents the number of sensors in the acquisition system, and $t_k$ is the sampling time. The $j$-th data in the preceding dataset window with missing values is taken as $\overline{D}_{W_{md,j}}^{t_k} = \{\overline{d}_{1,j}^{t_k}, \overline{d}_{2,j}^{t_k}, \cdots\cdots, \overline{d}_{n,j}^{t_k}\}$, where $\overline{d}_{n,j}^{t_k}$ is the data mean value of the $n$-th sensor in the preceding data set. The $j$-th data in the subsequent dataset window with missing values is taken as $\overline{\overline{D}}_{W_{md,j}} = \{\overline{\overline{d}}_{1,j}^{t_k}, \overline{\overline{d}}_{2,j}^{t_k}, \cdots\cdots, \overline{\overline{d}}_{n,j}^{t_k}\}$, where $\overline{\overline{d}}_{n,j}^{t_k}$ denotes the data mean value of the $n$-th sensor in the subsequent data set. Equation (17) is then applied to fill the missing data set:

$$d_i^{t_k} = \alpha \overline{d}_i^{t_k} + (1-\alpha)\overline{\overline{d}}_i^{t_k} \tag{17}$$

Equation (17) adopts a linear combination method, which comprehensively considers the correlation between the preceding and subsequent data and the missing data to perform linear filling on the missing data. In this equation, $\alpha$ denotes the reference value for linear combination with a value range of [0, 1]. The value of $\alpha$ is set to 1 when the missing bridge data are collected at the initial operation stage of the bridge (on the assumption that there is no structural deterioration in the initial stage), and the values at other sampling time are calculated in accordance with Equation (18):

$$\alpha = \frac{S(\theta)}{95} \tag{18}$$

Equation (18) indicates that $\alpha$ decreases with increased operational time. For long-term missing data, its filled value is closer to the subsequent data, thus reflecting the most unfavorable state of bridge resilience control.

## 5. Simulation analysis

### 5.1 Compression based on knowledge

In the simulation analysis, 50 cross-sectional dimensions were randomly selected as the validation set from the dataset of 3965 doubly symmetric I-shaped cross-sections (with flange thickness ranging from 700 mm to 1000 mm, web height ranging from 1200 mm to 1600 mm, and web thickness of 80 mm). Among the remaining cross-sections, 80% were used as the training set and the other 20% as the test set. A neural network model was then adopted for calculation, and the predicted compressive values and actual values of the 50 cross-sections outputted by the model are shown in Fig 1 ($I_x$) and Fig 2 ($I_y$) respectively.

The moment of inertia of the cross-section about the x-axis is presented in Fig 1, for which the maximum relative error of the predicted compressive values relative to the actual values is 7.03%, with a mean relative error of 2.42%. As shown in Fig 2 for the moment of inertia about the y-axis, the maximum and average relative errors are 4.54% and 1.28%, respectively.

### 5.2 Time-series data compression

Taking a simply supported beam with a reinforced concrete T-shaped cross-section as an example, C50 concrete was adopted for the beam, with a calculated span of 15 m, a flange width of 800 mm, a flange thickness of 100 mm, a cross-section height of 1200 mm and a web thickness of

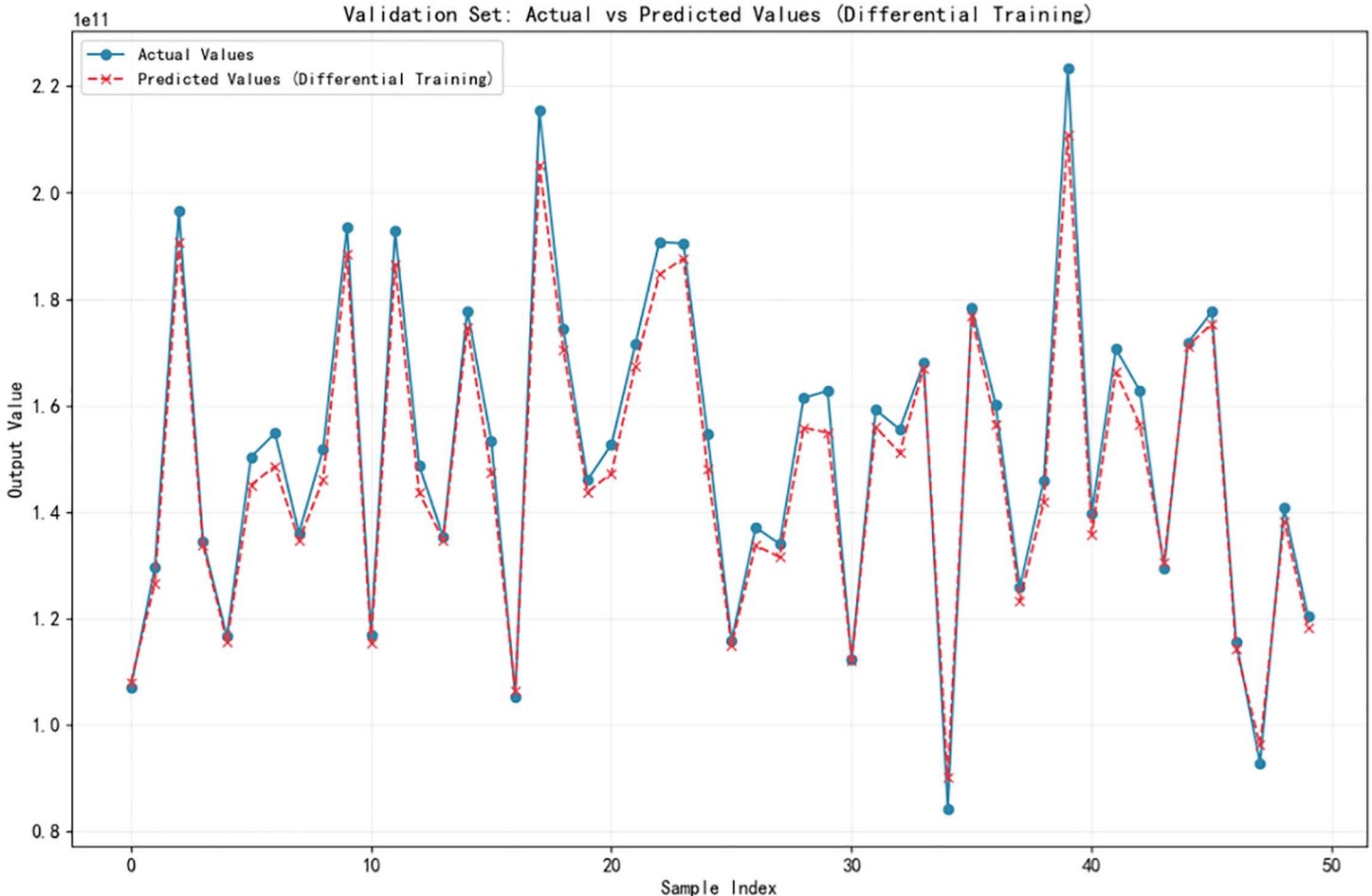

**Fig 1. Comparison between predicted compressive values and actual values of I$_x$ for doubly symmetric I-shaped cross-section.**

100 mm. 8% non-Gaussian noise (including traffic noise, wind noise and temperature noise) was added to the simulated acceleration data. The time-series data compression algorithm proposed in Section 4.2.2 was applied, and the corresponding compression results are presented in Fig 3.

As can be seen from Fig 3, the number of rows of the original data is 3600, and the number of rows of the data after compression is 262, with a compression ratio of 92.72%. The average data fidelity after compression is 97.77%, and the minimum data fidelity is 95.00%. To verify the advantages of the algorithm proposed in this paper compared with the existing algorithms, a comparative study was conducted between the proposed algorithm and the conventional algorithms (PCA, 1D Convolutional Auto-Encoder, Wavelet Transform, SAX and PAA) in the simulation analysis. Under the condition of 95% accuracy, the compression ratios of various algorithms are presented in Fig 4.

As can be seen from Fig 4, the compression ratio of the proposed algorithm in this paper is the highest under the premise of the same accuracy. Meanwhile, the compression ratios of the PAA and SAX algorithms are close to those of the time-series data compression algorithm proposed in this paper. However, PAA and SAX have inherent defects when applied to the field of bridge resilience control. Specifically, the PAA algorithm divides time-series data into fixed segments of equal length and uses the mean value of each segment to represent the characteristics of the entire segment. If the

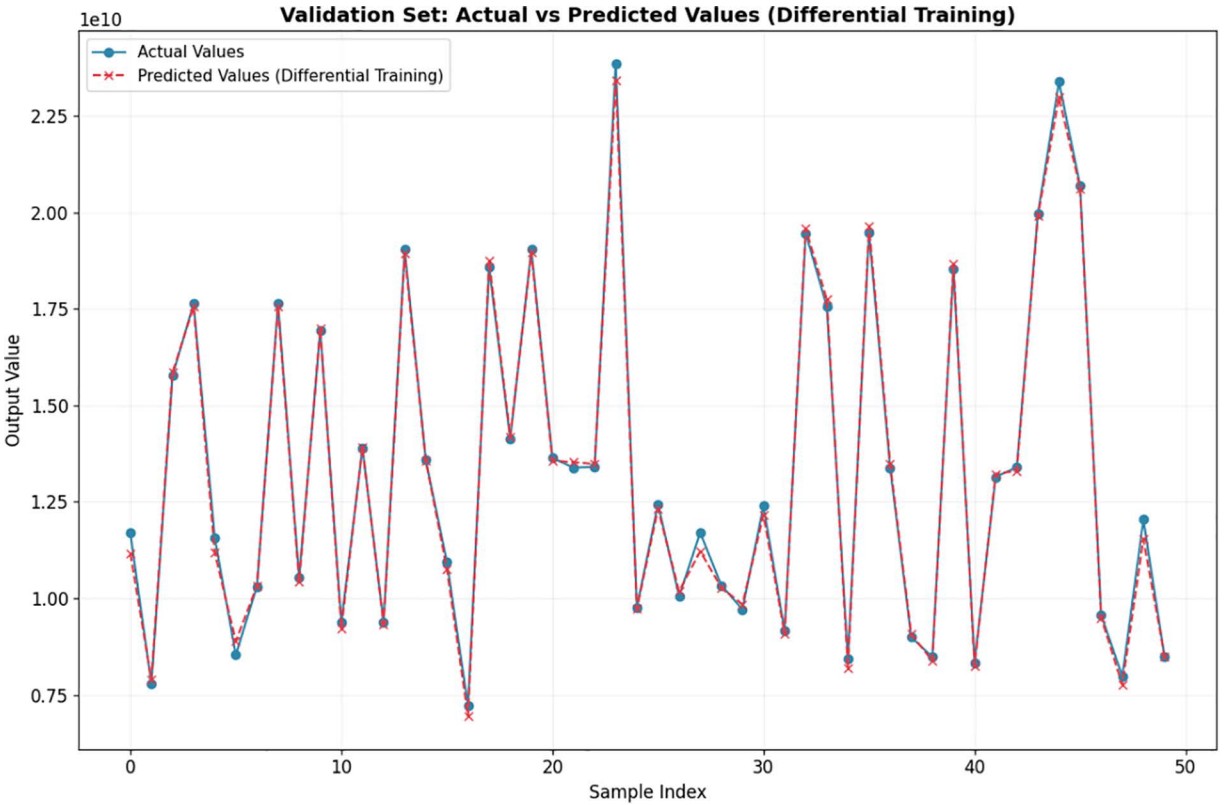

**Fig 2. Comparison between predicted compressive values and actual values of I$_y$ for doubly symmetric I-shaped cross-section.**

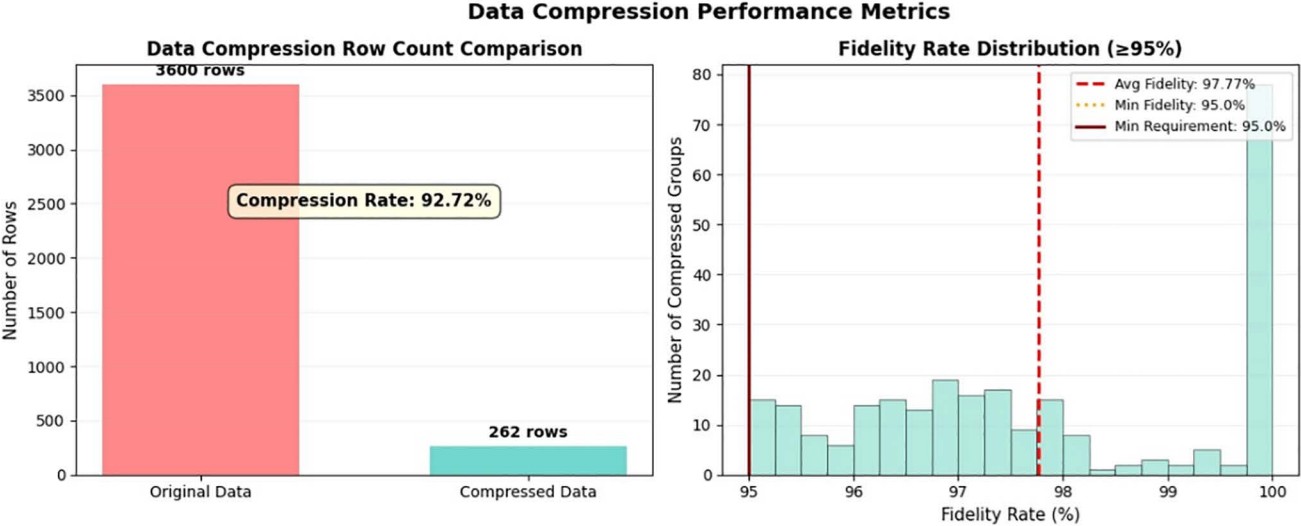

**Fig 3. Results of time-series data compression.**

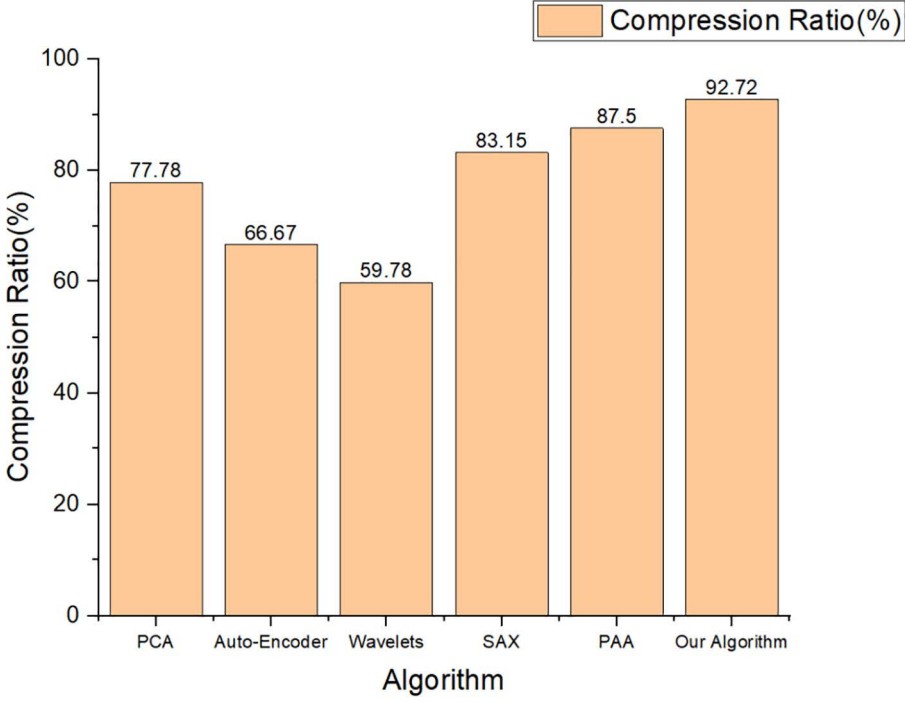

**Fig 4. Compression ratios of various algorithms (95% accuracy).**

peak value of bridge acceleration falls exactly on the segment boundary, this peak value will be averaged with other data in the segment, resulting in the loss of key signals of structural anomalies. The SAX algorithm adds a process of symbol mapping on the basis of the PAA algorithm. The SAX algorithm not only inherits the defects of the PAA algorithm, but also has some new drawbacks. For example, the SAX algorithm divides the symbol boundaries through the quantiles of the normal distribution, and the boundaries are fixed. In addition, the SAX algorithm only symbolizes the PAA values of a single segment without considering the temporal correlation between segments. Similarly, PCA, Auto-Encoder and Wavelets do not incorporate the characteristics of bridge structures, which may lead to the loss of key structural information and render the dimensionality reduction results devoid of engineering significance.

### 5.3 Sparse data filling and compression

The same case as that of time-series data compression was adopted for sparse data compression. In this study, it is assumed that the modal displacement data of Node 5 are missing in the range of 10% to 70%. The data filling method proposed in Section 4.3 was applied, and the errors after filling are presented in Table 1. In the simulation analysis, the operational time of the bridge is set as 10 years, and the window size is taken as 90 according to equation (15).

As can be seen from Table 1, the maximum MSE and mean MSE after data filling are both relatively small for different values of data missing ratio, which reflects that the results of data filling are consistent with the true values of the data. It should be noted that the error will be significantly large when abrupt changes occur in the data with the simultaneous absence of this segment of missing data. When the missing ratio of sparse data is 30%, and the window sizes are set as 90 (10 years of operational time), 80 (16 years of operational time), 70 (20 years of operational time), 60 (24 years of operational time) and 50 (26 years of operational time), the corresponding data compression ratios are presented in Table 2.

**Table 1. Error statistics of sparse data after filling.**

| Data Missing Ratio(%) | Med AE | Mean MSE |
|---|---|---|
| 10% | 0.002786 | 0.006175 |
| 20% | 0.002437 | 0.006804 |
| 30% | 0.001909 | 0.006920 |
| 40% | 0.002630 | 0.006688 |
| 50% | 0.001832 | 0.006465 |
| 60% | 0.002343 | 0.006961 |
| 70% | 0.002350 | 0.006814 |

**Table 2. Data compression ratios.**

| Window Sizes | Compression Ratio (%) |
|---|---|
| 90 | 88.61 |
| 80 | 89.44 |
| 70 | 90.14 |
| 60 | 90.36 |
| 50 | 90.33 |

As can be seen from Table 2, within the range of window sizes from 50 to 90, the data compression ratio is close to 90%, and there is little variation in the compression ratio across different window sizes. This phenomenon indicates that the result of data padding is actually a weighted average of the preceding and following data. The maximum difference between the compression ratio after padding and that of the original data is 4.4%..

## 6. Conclusion and outlook

Data is the core of bridge resilience control. However, with the continuous development of bridge resilience control technologies, the generation of massive data has posed new challenges to data storage, transmission and processing, resulting in low efficiency of bridge resilience control. To address this problem, domain knowledge, time-series data characteristics and bridge deterioration models are integrated into the data compression algorithm in this study, which achieves a substantial simplification of the data required for bridge resilience control. The main research conclusions are as follows:

First, domain knowledge is introduced into the general model, and a data compression algorithm based on domain knowledge is proposed. The results of simulation analysis show that the data compression ratio of this algorithm reaches 75%. Meanwhile, the compressed data has a clear physical meaning and can be directly applied to the resilience control of bridges.

Second, a compression algorithm combining time-series data and the bridge deterioration model is proposed. This algorithm realizes the refinement processing of dynamic time-series data, with the maximum refined compression ratio of the data reaching more than 92%. The results of comparative analysis with existing algorithms show that when the data fidelity rate is the same, the compression ratio of the algorithm proposed in this paper is higher, and the physical information of bridge structures can be retained effectively.

Third, a sparse data completion method based on the data sets before and after the measuring points is proposed. Meanwhile, the above-mentioned refined compression algorithm for time-series data is applied to compress the completed data, with the maximum data compression ratio after processing exceeding 90%.

The data compression algorithms proposed in this paper integrate the domain knowledge of bridge resilience control with the inherent time-series characteristics of data, which significantly reduces the volume of data required for bridge resilience control. Nevertheless, several issues in this study remain to be further explored and discussed:

First, the effectiveness of the algorithms is verified by simulated data in this study, and their actual performance will be validated in subsequent tests on real bridges. In particular, there are still differences between the simulated noise in the study and the actual noise of real bridges. Meanwhile, noise reduction algorithms will also be a key focus of future research.

Second, in the data compression based on domain knowledge, a biaxially symmetrical I-shaped section is adopted for analysis in this study. In follow-up research, analysis and comparison should be conducted for other types of cross-sections.

Third, the simply supported beam bridge is taken as an example to study the data compression algorithms in this research. Subsequent studies should carry out relevant analysis for other bridge types, such as continuous beam bridges, arch bridges, cable-stayed bridges and suspension bridges.

Fourth, the sparse data completion in this study only considers the data sets before and after a single measuring point. In the future, the research on sparse data completion for consecutive multiple measuring points will be carried out as a key focus.

## Nomenclature

| | |
|---|---|
| $d_i$ | Data |
| $l_i$ | data label |
| $t_i$ | Data type |
| $v_i$ | data value |
| $r_i$ | the associated information of the data |
| $k_i$ | the knowledge information that constrains the data |
| $w_i$ | data weight |
| $D$ | dataset |
| $kt_{i,k}$ | identifier for knowledge $k_i$ |
| $kr_{i,s}$ | $kr_{i,s}$ is the relationship between the value $v_i$ of data $d_i$ and the knowledge value $kv_{i,v}$ |
| $kv_{i,v}$ | the value of knowledge $k_i$ |
| $\bowtie$ | the knowledge association between data |
| $rt_i$ | the correlation type |
| $rp_i$ | the correlation parameter |
| $m_{i,j}$ | the key of the correlation |
| $p_{i,j}$ | the value of the correlation |
| $rt_t$ | the time of data acquisition |
| $y_i$ | the output of the neural network |
| $f(\zeta_i)$ | the activation function |
| $\omega_{ji}$ | the connection weight between the $j$th input neuron and the $i$-th output neuron |
| $x_{ji}$ | Input data of neurons |
| $e(\omega)$ | calculation error |
| $h_i$ | actual value |
| $\eta_{ji}$ | learning factor |
| $T$ | data template |
| $D_c$ | data loop |
| $A$ | displacement mode |
| $a_i$ | the displacement mode value analyzed based on the $i$-th sensor |

| $n$ | the number of sensors |
|---|---|
| $g_{k,i+z}$ | the displacement mode value at time $t_z$ |
| $W$ | window size |
| $\theta$ | the operational time of bridges |
| $S(\theta)$ | the bridge technical condition in the $\theta$-th year |
| $\varphi_{i-1}$ | the displacement mode at measurement point $i-1$ |
| $\varphi_i$ | the displacement mode at measurement point $i$ |
| $\varphi_{i+1}$ | the displacement mode at measurement point $i+1$ |
| $l$ | the distance between two measuring points |
| $\overline{\varphi}_i$ | the curvature mode at measurement point $i$ |
| $G$ | curvature mode diagram |
| $W_{md}$ | compensation window size |
| $D^{t_k}$ | missing data set |
| $\overline{D}^{t_k}_{W_{md},j}$ | The $j$-th data in the preceding dataset window with missing values |
| $\overline{\overline{D}}^{t_k}_{W_{md},j}$ | The $j$-th data in the subsequent dataset window with missing values |
| $d_i^{t_k}$ | the filled data of the $i$-th sensor |
| $\overline{d}^{t_k}_{n,j}$ | the data mean value of the $n$-th sensor in the preceding data set |
| $\overline{\overline{d}}^{t_k}_{n,j}$ | the data mean value of the $n$-th sensor in the subsequent data set |
| $\alpha$ | filling factor |

## Author contributions

**Conceptualization:** Ming Chen.

**Data curation:** Ming Chen.

**Formal analysis:** Ming Chen.

**Funding acquisition:** Ming Chen.

**Investigation:** Ming Chen.

**Methodology:** Ming Chen.

**Project administration:** Ming Chen.

**Resources:** Ming Chen.

**Software:** Ming Chen.

**Supervision:** Ming Chen.

**Validation:** Ming Chen.

**Visualization:** Ming Chen.

**Writing – original draft:** Ming Chen.

**Writing – review & editing:** Ming Chen.

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
