## [Decision Letter · Decision Letter 0]

24 Nov 2025

Dear Dr. Chen,

Improve the literature review and make it more focusedEnsure that all mathematical equations are correctly defined and are clear to readersProvide a detailed comparison with the state-of-the-art algorithms in the literature to provide validationPlease add a sensitivity analysis section to demonstrate the robustnessProvide a table to list every symbol/parameter in your workClosely follow the suggestions provided by reviewers to improve the revised version

plosone@plos.org. . . . A rebuttal letter that responds to each point raised by the academic editor and reviewer(s). You should upload this letter as a separate file labeled 'Response to Reviewers'.A marked-up copy of your manuscript that highlights changes made to the original version. You should upload this as a separate file labeled 'Revised Manuscript with Track Changes'.An unmarked version of your revised paper without tracked changes. You should upload this as a separate file labeled 'Manuscript'.

We look forward to receiving your revised manuscript.

Kind regards,

Babak Aslani, Ph.D.

Academic Editor

PLOS ONE

Journal Requirements:

https://journals.plos.org/plosone/s/file?id=ba62/PLOSOne_formatting_sample_title_authors_affiliations.pdf....

4. In the online submission form you indicate that your data is not available for proprietary reasons and have provided a contact point for accessing this data. Please note that your current contact point is a co-author on this manuscript. According to our Data Policy, the contact point must not be an author on the manuscript and must be an institutional contact, ideally not an individual. Please revise your data statement to a non-author institutional point of contact, such as a data access or ethics committee, and send this to us via return email. Please also include contact information for the third party organization, and please include the full citation of where the data can be found.

Reviewers' comments:

Reviewer's Responses to Questions

**Comments to the Author**

1. Is the manuscript technically sound, and do the data support the conclusions?

Reviewer #1: Yes

Reviewer #2: Partly

Reviewer #3: Yes

2. Has the statistical analysis been performed appropriately and rigorously?

Reviewer #1: Yes

Reviewer #2: Yes

Reviewer #3: Yes

3. Have the authors made all data underlying the findings in their manuscript fully available?

Reviewer #1: Yes

Reviewer #2: Yes

Reviewer #3: Yes

4. Is the manuscript presented in an intelligible fashion and written in standard English?

Reviewer #1: Yes

Reviewer #2: No

Reviewer #3: Yes

Reviewer #1: Overall Assessment

This paper addresses a highly relevant problem in bridge monitoring—the refinement of massive datasets for efficient use in large-scale models for resilience control. The motivation is clear and well-justified: as bridge monitoring systems become more sophisticated and generate increasingly large volumes of data, the need for intelligent data reduction strategies becomes critical. The authors demonstrate solid understanding of both structural engineering principles and computational methods, and their effort to integrate domain knowledge from bridge engineering into data processing algorithms represents a valuable research direction. The mathematical formalization of the problem shows rigor and careful thinking about how to represent bridge data systematically. The three-pronged approach—knowledge-based refinement, time-domain refinement, and sparse data handling—is comprehensive and addresses different aspects of the data management challenge. The simulation results, while preliminary, show promising refinement rates for time-domain data, which could have significant practical implications if validated.

Critical Issues Requiring Immediate Attention

1. Literature Review (Section 1.2)

The literature review is too broad and unfocused. Many references (e.g., [8] rumor detection, [12] text-to-image generation, [14] art generation) are tangentially related at best. Reduce this section and focus exclusively on: (a) data processing in structural health monitoring, (b) machine learning in bridge engineering, and (c) time-series compression methods. Clearly identify the research gap your work fills.

2. Lack of Validation and Comparison

This is the most serious deficiency. The paper presents NO comparisons with existing methods (PCA, autoencoders, wavelets, SAX, PAA) and uses ONLY synthetic data.

3. Methodological Clarity

Several methodological elements need clarification:

• Equation (10): The notation d(1-e^ξ/(1+e^ξ))/d(ξ) is ambiguous. Write it clearly as a derivative.

• Algorithm on page 13: "Apply a large-scale model to solve the data loop Dc…" is too vague. Specify the model architecture, training procedure, and loss function.

• Parameters like ε=0.05 and window size W need justification through sensitivity analysis.

4. Reproducibility

The statement "available upon reasonable request" is unacceptable for computational work in 2025. Create a public GitHub repository or archive it on Zenodo for a permanent DOI.

5. Results and Analysis

Current results are insufficient:

• Figure 1: Only 5 I-beam sections—need at least 50 varied cases

• Section 4.2: Only one simple T-beam. Test on continuous spans, cable-stayed, and arch bridges

• 5% Gaussian noise is unrealistic. Real bridges experience non-Gaussian noise from traffic, wind, and thermal effects

• No analysis of whether 97% compression preserves damage detection capability

6. Terminology Issues

The term "large-scale model" is used inconsistently throughout the manuscript, creating confusion about what type of computational approach is actually being employed. It is often unclear whether you are referring to Large Language Models such as GPT-like systems, deep learning models in general, or simply large neural networks with many parameters. This ambiguity is problematic because these are fundamentally different types of models with different architectures, training paradigms, and appropriate applications. To resolve this issue, you should use standard terminology from the machine learning and structural engineering literature. Specifically, use "deep learning models" when referring to multilayer neural networks, "neural networks" or "feedforward networks" for simpler architectures, and "convolutional networks" or "recurrent networks" when those specific architectures are employed. The term "large language models" or "LLMs" should be reserved exclusively for those instances where you are using transformer-based language models like GPT, BERT, or similar architectures, which appears to occur primarily in your graph classification step. This terminological precision will make your methodology clearer to readers and align your work with standard conventions in both the civil engineering and machine learning communities.

Essential Additions

The manuscript requires several critical additions to meet publication standards. First, you must add a comprehensive section titled "Comparison with State-of-the-Art" that includes a detailed table comparing your method against established techniques. This comparison table should include key metrics as well.

A second essential addition is a "Sensitivity Analysis" section that demonstrates the robustness of your proposed method across various conditions. This analysis should systematically vary noise levels different thresholds to show how the algorithm performs under different data quality scenarios. Additionally, you should test the method with missing data percentages ranging from 10% to 70% to establish the limits of your sparse data refinement approach. The sensitivity analysis must also include different bridge types. Finally, explore how variations in window size W and threshold ε affect the refinement rate and accuracy to guide future users in parameter selection.

Third, you must add a "Limitations" section that honestly discusses the constraints and assumptions of your method. This should explicitly acknowledge that the method assumes stationary Gaussian noise and regular sensor grids, which may not hold in all real-world scenarios. You must also acknowledge that validation has been limited to synthetic data and that the method's performance for subtle damage detection in early stages of deterioration remains unknown.

The abstract and contributions sections also require substantial improvement. The abstract should begin by clearly establishing the problem, such as "Bridge monitoring generates massive datasets that challenge current data management and analysis capabilities," before introducing your solution. You must clearly state what is genuinely novel rather than simply stating that you "propose" something. The percentages of 80% and 97% refinement rates need proper context explaining what these numbers mean for practical bridge management. Finally, you must distinguish clearly between your novel contributions, such as the specific application to bridges and the integration of deterioration models into the refinement algorithm, and your adaptations of existing techniques like windowing and interpolation methods.

Language and Presentation

The manuscript requires professional English editing to meet publication standards. Throughout the text, there are numerous instances of awkward phrasing that suggest direct translation from another language. For example, phrases like "with the increase of bridge operation time" should be simplified to "with increased operational time," and expressions such as "cannot be simply integrated" would read more naturally as "cannot be directly unified." Beyond these specific examples, the manuscript suffers from inconsistent mathematical notation, with variables sometimes written as d_i and other times as d_{i,j} without clear explanation of when each form is appropriate, and approximation symbols alternating between ≈ and ≃ apparently interchangeably. These inconsistencies, while seemingly minor, can confuse readers and detract from the technical content.

Reviewer #2: This work examines data preprocessing for bridge-resilience control supported by large models. As bridge-resilience control technology advances, the sheer volume of data has begun to undermine control efficiency. To address this issue, the author proposed a preprocessing method that incorporates domain knowledge and time-domain features, and demonstrated its use in simulation analyses. The problem tackled is worthwhile; however, the manuscript does not yet demonstrate sufficient scientific rigor or value and would require major revision before it could be considered for publication. The specific modification suggestions are as follows:

1. Please restructure the Introduction according to the standard logic of the research paper; the current logic does not make the theoretical or technical contribution clear.

2. Move the literature review currently in Section 1.2 to a new '2. Related Work' section.

3. There are a large number of inaccurate paragraph divisions in the manuscript. For example,

“Correlation refers to the correlation between data. The correlations studied in this article include geometric correlations and temporal correlations.

Geometric correlation refers to the geometric shapes or structures composed of data within the subset D_k of a dataset, D_k\subset D. When D_k==D , D_k constitutes the entire bridge structure. Time domain correlation refers to the correlation generated in the time domain of a certain data of a bridge structure over time. For example, the acceleration data and displacement data collected by the bridge health monitoring system all satisfy time-domain correlation.”

should be a single paragraph. Please correct all such paragraph divisions throughout the manuscript.

4. Provide a nomenclature table that lists every symbol/parameter used in the paper.

5. All algorithm processes are currently presented in a non-standard format: they lack numbering, explicit Input/Output blocks, etc. In addition to presenting the algorithms in the standard way, it is recommended to use more descriptive language for the algorithm processes. For example, using '→' directly in an algorithm without explanation can confuse readers (in mathematical expression, '→' can mean 'tends to'; in pseudocode, it is often understood as 'assignment').

6. Supply pseudocode for every algorithm, either in the main text or in an appendix.

7. Step 7 of the time domain data refinement algorithm states 'Call the large-scale model to classify G and generate a classification quantity k'. What the large-scale model is used in this step? What is the specific classification algorithm? How are the specific parameters set? If these issues are not clearly stated in the manuscript, the scientificity and accuracy of the subsequent analysis results cannot be guaranteed.

8. The claim 'The resilience-control data of bridges is affected by factors such as the number of equipment and equipment failures' is unconvincing as an explanation for data sparsity. Commercial sensing systems are designed to fail rarely; therefore, device failure alone cannot justify the observed scarcity. Provide stronger evidence or cite published studies that substantiate this argument.

9. Justify the choice of \varepsilon=0.05 in Equation (18).

10. Compare the proposed refinement algorithms with state-of-the-art alternatives to demonstrate their superiority.

Reviewer #3: Comments to Author(s)

Article Title: The title is very ok and it flows along just that it is too long the author should find away to refine it abit.

Abstract

The abstract is good to go; everything is stated correctly, from its introduction to the problem, methodology, and result output was mentioned as related to its simulation analysis but it does not mention its performance explicitly rather it proceeded to the future research.

It is also advisable to use a grammar editor or an expert in grammar or language to assist in correcting grammatical and punctuation errors.

LITERATURE REVIEW: The review is done, properly referenced, concurrence is important as it is supposed to flow along and tailored to the current research which means that the closest research need to be considered.

- PERFOMANCE METRICS

o Algorithms are well and properly stated, The system model and the transformation of time domain data is well articulated.

METHODOLOGY:

- This is properly presented, there are a lot of information mentioned in the methods and strategy which has been explicitly described.

- Kindly re-arrange the Methodology for simplicity and easy tracking so that It can be simpler to relate the methods, results and output.

RESULT

- The result is well presented inclusive of all necessary graph.

- The concluding part is too scanty compare to too many equations and solutions provided, I will suggest it should be a more explicit way of concluding

- Supporting information is not necessary, but it can also be integrated into the conclusion in a summarized manner.

CONCLUSION:

This paper is good as it stated the key aspect of the research and it widen the scope as it continue to state explicitly the research motivation, and it is very important in the core area of Algorithm and Case Analysis so it requires a lot of attention while dissemination them.

.

Reviewer #1: No

Reviewer #2: No

Reviewer #3: No

---

## [Author Response · Author response to Decision Letter 1]

30 Dec 2025

Dear Dr Babak Aslani and Reviewers,

Thank you very much for taking the time to review my manuscript entitled " Data Refinement Processing of Bridge Resilience Control Supported by Large-scale model Algorithm and Case Analysis". I greatly appreciate the constructive feedback and insightful comments, which have significantly helped me improve the quality of my work. Below, I provide a point-by-point response to the reviewers' comments. Below, I provide a detailed response to each comment.

Reviewer 1

Comment 1: Literature Review (Section 1.2)

The literature review is too broad and unfocused. Many references (e.g., [8] rumor detection, [12] text-to-image generation, [14] art generation) are tangentially related at best. Reduce this section and focus exclusively on: (a) data processing in structural health monitoring, (b) machine learning in bridge engineering, and (c) time-series compression methods. Clearly identify the research gap your work fills.

Response 1:

According to the reviewer's comments, the Literature Review has been revised by reorganizing the references based on (a) data processing in structural health monitoring, (b) machine learning in bridge engineering, and (c) time series compression methods.

Comment 2: Lack of Validation and Comparison

This is the most serious deficiency. The paper presents NO comparisons with existing methods (PCA, autoencoders, wavelets, SAX, PAA) and uses ONLY synthetic data.

Response 2:

Following the comments from the reviewers, the proposed time-series data compression algorithm is compared with mainstream dimensionality reduction and compression algorithms, including PCA, Auto-Encoders, Wavelet Transform, SAX and PAA. Specifically, the 1D Convolutional Auto-Encoder was employed as the Auto-Encoder method in the comparison. The corresponding comparison results are presented in Fig. 4 of the manuscript.

Comment 3：Methodological Clarity

Several methodological elements need clarification:

3.1 Equation (10): The notation d(1-e^ξ/(1+e^ξ))/d(ξ) is ambiguous. Write it clearly as a derivative.

Algorithm on page 13: "Apply a large-scale model to solve the data loop Dc…" is too vague. Specify the model architecture, training procedure, and loss function.

Parameters like ε=0.05 and window size W need justification through sensitivity analysis.

Response 3:

In accordance with the reviewers' comments, Eq. (10) is revised into the following form. Meanwhile, in Eq. (11), d_j→w_j is modified to d_j (w_j).

η_ji=-0.2×d/dξ ((1-e^ξ)/(1+e^ξ )) (10)

〖ξ=1-(d〗_j (w_j)) (11)

In the meantime, Equation (9) has been modified to the following form to adopt the differential learning rate.

ω_ji=ω_ji-η_ji x_ji (y_i-d_i) (9)

In accordance with the reviewers' comments, we have specified which large-scale model to invoke, and revised the expression "Apply a large-scale model to solve the data loop Dc…" to "Apply the neural network model to solve the data loop Dc…". Meanwhile, a brief introduction to the architecture, training procedure, and loss function of the neural network model is added.

In accordance with the reviewers' guidance, the window size is set in a dynamic manner, and the detailed revisions can be found in Section 4.2.1 of the manuscript. In addition, ε has been removed in the revised manuscript, and the value of α is determined according to Eq. (18) in the revised manuscript.

Comment 4：Reproducibility

The statement "available upon reasonable request" is unacceptable for computational work in 2025. Create a public GitHub repository or archive it on Zenodo for a permanent DOI.

Response4:

In accordance with the reviewers' comments, the dataset and source code have been made publicly available.

Comment 5：Results and Analysis

Current results are insufficient:

Figure 1: Only 5 I-beam sections—need at least 50 varied cases

Section 4.2: Only one simple T-beam. Test on continuous spans, cable-stayed, and arch bridges

5% Gaussian noise is unrealistic. Real bridges experience non-Gaussian noise from traffic, wind, and thermal effects。

No analysis of whether 97% compression preserves damage detection capability

Response 5:

In accordance with the reviewers' comments, fifty cross-sections are presented in Fig. 1.

The reviewers suggested conducting tests on different bridge types including continuous span bridges, cable-stayed bridges and arch bridges. We are currently conducting relevant research, and this work will be completed in our subsequent papers.

In accordance with the reviewers' comments, the noise has been modified to 8% non-Gaussian noise (including vehicle noise, wind noise and temperature noise) in the revised manuscript.

In addition, due to the introduction of noise, the compression ratio of the time series data compression algorithm proposed in the revised manuscript is 89.42%. We will analyze the influence of the compressed data on bridge damage detection in subsequent research.

Comment 6：

Terminology Issues

The term "large-scale model" is used inconsistently throughout the manuscript, creating confusion about what type of computational approach is actually being employed. It is often unclear whether you are referring to Large Language Models such as GPT-like systems, deep learning models in general, or simply large neural networks with many parameters. This ambiguity is problematic because these are fundamentally different types of models with different architectures, training paradigms, and appropriate applications. To resolve this issue, you should use standard terminology from the machine learning and structural engineering literature. Specifically, use "deep learning models" when referring to multilayer neural networks, "neural networks" or "feedforward networks" for simpler architectures, and "convolutional networks" or "recurrent networks" when those specific architectures are employed. The term "large language models" or "LLMs" should be reserved exclusively for those instances where you are using transformer-based language models like GPT, BERT, or similar architectures, which appears to occur primarily in your graph classification step. This terminological precision will make your methodology clearer to readers and align your work with standard conventions in both the civil engineering and machine learning communities.

Essential Additions

The manuscript requires several critical additions to meet publication standards. First, you must add a comprehensive section titled "Comparison with State-of-the-Art" that includes a detailed table comparing your method against established techniques. This comparison table should include key metrics as well. A second essential addition is a "Sensitivity Analysis" section that demonstrates the robustness of your proposed method across various conditions. This analysis should systematically vary noise levels different thresholds to show how the algorithm performs under different data quality scenarios. Additionally, you should test the method with missing data percentages ranging from 10% to 70% to establish the limits of your sparse data refinement approach. The sensitivity analysis must also include different bridge types. Finally, explore how variations in window size W and threshold ε affect the refinement rate and accuracy to guide future users in parameter selection. Third, you must add a "Limitations" section that honestly discusses the constraints and assumptions of your method. This should explicitly acknowledge that the method assumes stationary Gaussian noise and regular sensor grids, which may not hold in all real-world scenarios. You must also acknowledge that validation has been limited to synthetic data and that the method's performance for subtle damage detection in early stages of deterioration remains unknown.

The abstract and contributions sections also require substantial improvement. The abstract should begin by clearly establishing the problem, such as "Bridge monitoring generates massive datasets that challenge current data management and analysis capabilities," before introducing your solution. You must clearly state what is genuinely novel rather than simply stating that you "propose" something. The percentages of 80% and 97% refinement rates need proper context explaining what these numbers mean for practical bridge management. Finally, you must distinguish clearly between your novel contributions, such as the specific application to bridges and the integration of deterioration models into the refinement algorithm, and your adaptations of existing techniques like windowing and interpolation methods.

Language and Presentation

The manuscript requires professional English editing to meet publication standards. Throughout the text, there are numerous instances of awkward phrasing that suggest direct translation from another language. For example, phrases like "with the increase of bridge operation time" should be simplified to "with increased operational time," and expressions such as "cannot be simply integrated" would read more naturally as "cannot be directly unified." Beyond these specific examples, the manuscript suffers from inconsistent mathematical notation, with variables sometimes written as d_i and other times as d_{i,j} without clear explanation of when each form is appropriate, and approximation symbols alternating between ≈ and ≃ apparently interchangeably. These inconsistencies, while seemingly minor, can confuse readers and detract from the technical content.

Response 6:

In accordance with the reviewers' comments, the term "large model" has been revised to "neural network model" in the revised manuscript, so that readers can clearly understand the method adopted in this study.

The comparison of compression ratios between the time-series data compression algorithm proposed in this paper and the existing algorithms has been supplemented in the revised manuscript. These comparisons are based on the premise that all algorithms adopt the same data fidelity rate (95%).

For missing data compression, the results of data completion for data missing rates ranging from 10% to 70% have been supplemented (Table 1). Meanwhile, taking the 30% data missing rate as an example, the compression ratios corresponding to the initial values of different window sizes have been calculated (Table 2).

In accordance with the reviewers' comments, the limitations of this study have been added to the Conclusions and Future Work section. In particular, it is stated explicitly that the verification is completed based on simulated data, and the actual performance of the proposed algorithms will be further validated in subsequent tests on real bridges.

In accordance with the reviewers' comments, the application value of the compression ratio and data fidelity rate for bridge resilience control has been clearly specified in the abstract.

The expressions and symbols in the manuscript have been revised in accordance with the reviewers' comments.

Reviewer 2

Comment 1:

Please restructure the Introduction according to the standard logic of the research paper; the current logic does not make the theoretical or technical contribution clear.

Response 1: In accordance with the reviewers' comments, the introduction has been restructured, and the Motivation and Innovation section has been revised.

Comment 2:

Move the literature review currently in Section 1.2 to a new '2. Related Work' section.

Response 2: In accordance with the reviewers' comments, Section 1.2 has been moved to Section 2 "Related Work".

Comment 3:

There are a large number of inaccurate paragraph divisions in the manuscript. For example, “Correlation refers to the correlation between data. The correlations studied in this article include geometric correlations and temporal correlations. Geometric correlation refers to the geometric shapes or structures composed of data within the subset D_k of a dataset, D_k\subset D. When D_k==D , D_k constitutes the entire bridge structure. Time domain correlation refers to the correlation generated in the time domain of a certain data of a bridge structure over time. For example, the acceleration data and displacement data collected by the bridge health monitoring system all satisfy time-domain correlation.” should be a single paragraph. Please correct all such paragraph divisions throughout the manuscript.

Response 3:

In accordance with the reviewers' comments, the paragraph division of the entire manuscript has been revised in the revised version.

Comment 4:

Provide a nomenclature table that lists every symbol/parameter used in the paper.

Response 4:

A nomenclature of all symbols has been added to Appendix A of the revised manuscript.

Comment 5:

All algorithm processes are currently presented in a non-standard format: they lack numbering, explicit Input/Output blocks, etc. In addition to presenting the algorithms in the standard way, it is recommended to use more descriptive language for the algorithm processes. For example, using '→' directly in an algorithm without explanation can confuse readers (in mathematical expression, '→' can mean 'tends to'; in pseudocode, it is often understood as 'assignment').

Response 5:

In the revised manuscript, we have presented the original algorithm in the form of pseudocode to enable readers to understand it more clearly.

Comment 6:

Supply pseudocode for every algorithm, either in the main text or in an appendix.

Response 6:

In accordance with the reviewers' requirements, we have provided the pseudocode of the algorithm in the revised manuscript.

Comment 7:

Step 7 of the time domain data refinement algorithm states 'Call the large-scale model to classify G and generate a classification quantity k'. What the large-scale model is used in this step? What is the specific classification algorithm? How are the specific parameters set? If these issues are not clearly stated in the manuscript, the scientificity and accuracy of the subsequent analysis results cannot be guaranteed.

Response 7:

In Step 7，the DeepSeek-chat model is invoked to classify the graphs generated by the curvature mode. When the number of categories X=1, the data in this category is compressed by taking the mean value of the data within the category. When the number of categories X>1, half of the window size is used to perform reclassification. Meanwhile, the data fidelity rate is set to 95% in the program design. In the revised manuscript, the large model is specified as DeepSeek-chat.

Comment 8:

The claim 'The resilience-control data of bridges is affected by factors such as the number of equipment and equipment failures' is unconvincing as an explanation for data sparsity. Commercial sensing systems are designed to fail rarely; therefore, device failure alone cannot justify the observed scarcity. Provide stronger evidence or cite published studies that substantiate this argument.

Response 8:

In response to the reviewers' comments, we have cited references [24, 25, 26] in the revised manuscript to demonstrate that the sparse nature of the bridge resilience control data arises from factors such as the number of devices and device failures.

Comment 9:

Justify the choice of ε=0.05 in Equation (18).

Response 9:

In the revised manuscript, ε has been removed in the revised manuscript, and the value of α is determined according to Eq. (18) in the revised manuscript.

Comment 10:

Compare the proposed refinement algorithms with state-of-the-art alternatives to demonstrate their superiority.

Response 10:

Following the comments from the reviewers, the proposed time-series data compression algorithm is compared with mainstream dimensionality reduction and compression algorithms, including PCA, Auto-Encoders, Wavelet Transform, SAX and PAA. Specifically, the 1D Convolutional Auto-Encoder was employed as the Auto-Encoder method in the comparison. The corresponding comparison results are presented in Fig. 4 of the manuscript.

Reviewer 3

Comment 1:

Abstract

The abstract is good to go; everything is stated correctly, from its introduction to the problem, methodology, and result output was mentioned as related to its simul

---

## [Decision Letter · Decision Letter 1]

26 Jan 2026

Dear Dr. Chen,

Please closely read the reviewer feedback regarding using LLM for graph classification. I believe that this part of your work can be revised with a more solid choice. So, please make sure to address this critical point in the revised version.

plosone@plos.org. . . . A letter that responds to each point raised by the academic editor and reviewer(s). You should upload this letter as a separate file labeled 'Response to Reviewers'.A marked-up copy of your manuscript that highlights changes made to the original version. You should upload this as a separate file labeled 'Revised Manuscript with Track Changes'.An unmarked version of your revised paper without tracked changes. You should upload this as a separate file labeled 'Manuscript'.

We look forward to receiving your revised manuscript.

Kind regards,

Babak Aslani

Academic Editor

PLOS One

Journal Requirements:

Reviewers' comments:

Reviewer's Responses to Questions

**Comments to the Author**

Reviewer #1: All comments have been addressed

Reviewer #2: All comments have been addressed

2. Is the manuscript technically sound, and do the data support the conclusions?

Reviewer #1: Yes

Reviewer #2: No

3. Has the statistical analysis been performed appropriately and rigorously?

Reviewer #1: Yes

Reviewer #2: Yes

4. Have the authors made all data underlying the findings in their manuscript fully available?

Reviewer #1: Yes

Reviewer #2: Yes

5. Is the manuscript presented in an intelligible fashion and written in standard English?

Reviewer #1: Yes

Reviewer #2: Yes

Reviewer #1: The author has addressed all the reviewer concerns, therefore he has fullfilled all the requirements.

Reviewer #2: The revised manuscript presents its innovation and technical details more clearly, but also exposes potential technical risks. The DeepSeek-chat model, a famous LLM, is invoked to classify the graphs generated by the curvature mod. However, LLM is probability generated. Its primary goal is to generate natural, diverse, and creative text, rather than conducting stable deterministic calculations. For tasks such as graph classification that require structural awareness and deterministic computation, specialized GNN models far outperform LLM in accuracy, efficiency, consistency, and interpretability. LLM is more suitable as an auxiliary tool for interpreting, summarizing, or generating descriptive text based on graph data. Therefore, I believe that there is technical uncertainty in this work, which may lead to potential technical risks in the actual operation process. My final suggestion is to reject the manuscript.

In addition, there is a small issue that the author can refer to: the next paragraph of equation (15) repeats the discussion after equation (14).

.

Reviewer #1: No

Reviewer #2: No

---

## [Author Response · Author response to Decision Letter 2]

2 Mar 2026

Dear Dr Babak Aslani and Reviewers,

Thank you very much for taking the time to review my manuscript entitled " Data Refinement Processing of Bridge Resilience Control Supported by Large-scale model Algorithm and Case Analysis". I greatly appreciate the constructive feedback and insightful comments, which have significantly helped me improve the quality of my work. Below, I provide a point-by-point response to the reviewers' comments.

Reviewer 2

Comment:

The revised manuscript presents its innovation and technical details more clearly, but also exposes potential technical risks. The DeepSeek-chat model, a famous LLM, is invoked to classify the graphs generated by the curvature mod. However, LLM is probability generated. Its primary goal is to generate natural, diverse, and creative text, rather than conducting stable deterministic calculations. For tasks such as graph classification that require structural awareness and deterministic computation, specialized GNN models far outperform LLM in accuracy, efficiency, consistency, and interpretability. LLM is more suitable as an auxiliary tool for interpreting, summarizing, or generating descriptive text based on graph data. Therefore, I believe that there is technical uncertainty in this work, which may lead to potential technical risks in the actual operation process. My final suggestion is to reject the manuscript.

In addition, there is a small issue that the author can refer to: the next paragraph of equation (15) repeats the discussion after equation (14).

Response :

I sincerely appreciate the insightful and constructive comments raised by the reviewer, which have identified a key technical limitation in my manuscript and provided valuable guidance for enhancing the rigor and reliability of the research. We fully recognize and accept the core viewpoints of the reviewer. In the revised manuscript, I have adopted the GIN (Graph Isomorphism Network) to judge the similarity between graphs, and revised the algorithm process and corresponding results accordingly. Meanwhile, the redundant content in the paragraphs following Formula (15) has been deleted.

Sincerely,

Ming Chen

Shanghai ZhongQiao Vocational and Technical University

chenmchen1975@126.com

---

## [Decision Letter · Decision Letter 2]

17 Mar 2026

Data compression of Bridge Resilience Control: Algorithm and Case

PONE-D-25-50186R2

Dear Dr. Chen,

We’re pleased to inform you that your manuscript has been judged scientifically suitable for publication and will be formally accepted for publication once it meets all outstanding technical requirements.

Kind regards,

Babak Aslani, Ph.D.

Academic Editor

PLOS One

Additional Editor Comments (optional):

Reviewers' comments:

Reviewer's Responses to Questions

**Comments to the Author**

Reviewer #1: All comments have been addressed

Reviewer #2: All comments have been addressed

2. Is the manuscript technically sound, and do the data support the conclusions?

Reviewer #1: Yes

Reviewer #2: Yes

3. Has the statistical analysis been performed appropriately and rigorously?

Reviewer #1: Yes

Reviewer #2: Yes

4. Have the authors made all data underlying the findings in their manuscript fully available?

Reviewer #1: Yes

Reviewer #2: Yes

5. Is the manuscript presented in an intelligible fashion and written in standard English?

Reviewer #1: Yes

Reviewer #2: Yes

Reviewer #1: Every comment was addressed succesfully by the author following the indications, therefore recommend its publication under its present form.

Reviewer #2: Thank you very much for the author's efforts in revising the manuscript and completing the experimental part again. The methods and techniques used in this manuscript are reasonable, and the quality of the manuscript has been greatly improved. I suggest accepting this manuscript for publication on PlOS One.

.

Reviewer #1: No

Reviewer #2: No

---

## [Editor Report · Acceptance letter]

PONE-D-25-50186R2

PLOS One

Dear Dr. Chen,

I'm pleased to inform you that your manuscript has been deemed suitable for publication in PLOS One. Congratulations! Your manuscript is now being handed over to our production team.

Kind regards,

on behalf of

Dr. Babak Aslani

Academic Editor

PLOS One